# Organic Nanoparticles Based on D-A-D Small Molecule: Self-Assembly, Photophysical Properties, and Synergistic Photodynamic/Photothermal Effects

**DOI:** 10.3390/ma15020502

**Published:** 2022-01-10

**Authors:** Liangliang Yue, Haolan Li, Qi Sun, Xiaogang Luo, Fengshou Wu, Xunjin Zhu

**Affiliations:** 1Hubei Key Laboratory of Novel Reactor and Green Chemical Technology, Key Laboratory for Green Chemical Process of the Ministry of Education, School of Chemical Engineering and Pharmacy, Wuhan Institute of Technology, Wuhan 430205, China; yuel1994@163.com (L.Y.); lhl970816@163.com (H.L.); xgluo@wit.edu.cn (X.L.); 2Department of Chemistry, Hong Kong Baptist University, Waterloo Road, Hong Kong, China; 3School of Chemistry and Environmental Engineering, Wuhan Institute of Technology, Wuhan 430205, China; qisun@wit.edu.cn; 4School of Materials Science and Engineering, Zhengzhou University, Zhengzhou 450001, China

**Keywords:** perylene diimide, self-assembly, organic nanomaterials, photothermal therapy, reactive oxygen species

## Abstract

Cancer is one of the major diseases threatening human health. Traditional cancer treatments have notable side-effects as they can damage the immune system. Recently, phototherapy, as a potential strategy for clinical cancer therapy, has received wide attention due to its minimal invasiveness and high efficiency. Herein, a small organic molecule (PTA) with a D-A-D structure was prepared via a Sonogashira coupling reaction between the electron-withdrawing dibromo-perylenediimide and electron-donating 4-ethynyl-*N*,*N*-diphenylaniline. The amphiphilic organic molecule was then transformed into nanoparticles (PTA-NPs) through the self-assembling method. Upon laser irradiation at 635 nm, PTA-NPs displayed a high photothermal conversion efficiency (PCE = 43%) together with efficient reactive oxygen species (ROS) generation. The fluorescence images also indicated the production of ROS in cancer cells with PTA-NPs. In addition, the biocompatibility and photocytotoxicity of PTA-NPs were evaluated by 3-(4,5-Dimethylthiazol-2-yl)-2,5-diphenyltetrazolium bromide (MTT) assay and live/dead cell co-staining test. Therefore, the as-prepared organic nanomaterials were demonstrated as promising nanomaterials for cancer phototherapy in the clinic.

## 1. Introduction

Because of its high mortality, cancer has always been a great threat to human health [1]. Today, a considerable number of traditional cancer treatments exist. However, most of these conventional cancer treatments cause irreversible side-effects to healthy cells in clinical application [2,3]. During the last decade, much effort has been dedicated to explore new cancer treatment methods with low side-effects [4,5] Photodynamic therapy (PDT) and photothermal therapy (PTT) showed the advantages of minimal invasiveness and high therapeutic efficiency, thus attracting much attention as a promising therapeutic technique [6,7,8,9]. In PDT, the light irradiation of photosensitizers produces reactive oxygen species (ROS) which can abolish cancer cells and tissue [10,11]. PTT mainly depends on the local heat effect induced by near-infrared light irradiation on photothermal agents, resulting in the damage of tumor cells [12,13].

Based on the mechanism of PDT and PTT, much effort has been dedicated to developing photosensitizers with high ROS generation or efficient photothermal conversion efficiency (PCE) for effective cancer phototherapy. Up to now, a great number of photosensitizers including organic and inorganic functional materials have been designed and studied for PTT and PDT, such as noble metal nanoparticles [14], metal chalcogenide nanomaterials [15], carbon-based nanomaterials [16], magnetic nanoparticles [17], and polymer materials [18]. Although inorganic materials generally exhibit high photothermal conversion efficiency or efficient ROS generation due to their semiconductor properties, they are limited in clinic application by their long-term toxicity toward organisms. Recently, organic materials with excellent biocompatibility have gained considerable interest for PTT or PDT applications, such as porphyrin [19,20], phthalocyanine [21,22], and diketopyrrolopyrrole [23,24]. In 2017, Dong et al. synthesized organic NPs from an organic molecule, which was constructed by conjugating triphenylamine (TPA) with 3,6-di(2-thienyl)-2,5-dihydropyrrolo[3,4-c]pyrrole-1,4-dione (DPP). The prepared organic materials showed efficient singlet oxygen (^1^O_2_) generation (Φ_Δ_ = 33.6%) and high PCE (34.5%) under 660 nm laser irradiation [25]. Yoon et al. developed phthalocyanine-based nanomaterials (NanoPcTBs) with both photothermal and photoacoustic properties [26]. In recent years, our group also developed a series of small molecule-based nanomaterials with strong absorption in the near-infrared region, which could act as efficient nanoagents in photodynamic or photothermal therapy [8,19,22].

Perylene diimide (PDI) derivatives have been widely studied for phototherapy for their high thermal stability, large π–π conjugated system, good photochemical properties, and bright luminescence. However, the absence of near-infrared (NIR) absorption and the extreme hydrophobic properties of PDI molecules inhibit their practical application in PTT [27,28]. Herein, we designed and synthesized a small organic molecule PTA with a D-A-D structure, where perylene diimide was used as an electronic accepter while triphenylamine was used as an electronic donor. The two components were conjugated through a triple bond, which enhanced the π conjugation of the system, leading to a red shift of the absorption spectrum [29]. In order to enhance its water solubility, polyethylene glycol chains were introduced onto the perylene diimide unit. The amphiphilic PTA was then self-assembled into the related nanostructures (PTA-NPs) through the nanoprecipitation method [30]. PTA-NPs exhibited high photothermal conversion efficiency (PCE = 43%) under 635 nm laser, which was comparable to that of reported organic nanomaterials (Table 1). The photocytotoxicity and intracellular ROS generation of PTA-NPs were finally verified by MTT assay and fluorescence images. Compared to those reported nanomaterials, the PTA-NPs showed several distinct properties, including (1) organic nanomaterials generated from small molecules (PTA) with a clear structure and accurately determined molecular weight, (2) high photothermal conversion efficiency (43%), and (3) simple fabrication method without the addition of any other reagents. Therefore, PTA-NPs are very promising for photothermal cancer therapy in preclinical applications.

## 2. Experimental Sections

### 2.1. Synthesis

*N*,*N*-Bis[2-(2-(2-Ethoxyethoxy)ethoxy)ethylane]-1,7-dibromoperylene-3,4,9,10-tetracarboxylic acid bisimide (10) [35], 4-bromo-*N*,*N*-diphenyl aniline (6) [36], and 4-ethynyl-*N*,*N*-diphenylaniline (7) [37] were prepared according to methods in the literature (Appendix A).

*N*,*N*-Bis[2-(2-(2-ethoxyethoxy)ethoxy)ethylane]-1,7-di(4-ethynyl-*N*,*N*-diphenyl aniline)-3,4,9,10-tetracarboxylic acid bisimide (PTA) [38].

A two-necked round-bottom flask was filled with Compound 10 (20.00 mg, 0.023 mmol) THF (10 mL), triethylamine (TEA) (8 mL), CuI (0.8 mg, 0.0042 mmol), and PdCl_2_(PPh_3_)_2_ (3 mg, 0.0042 mmol) under an N_2_ atmosphere. After the temperature was increased to 50 °C, Compound 7 (23.52 mg, 0.069 mmol) was added to the system slowly and then heated and kept at 70 °C for 6 h. The residue was obtained by solvent evaporation and then purified through column (silica gel) chromatography (DCM:CH_3_OH = 45:1) to afford the desired product as a purple-black solid (17.18 mg, 60%). ^1^H-NMR (400 MHz, CDCl_3_) δ 9.91 (d, *J* = 8.0 Hz, 2H), 8.65 (s, 2H), 8.47 (d, *J* = 8.4 Hz, 2H), 7.42 (d, *J* = 8.3 Hz, 4H), 7.33 (t, *J* = 7.5 Hz, 8H), 7.20–7.10 (m, 12H), 7.05 (d, *J* = 8.4 Hz, 4H), 4.45 (s, 4H), 3.84 (s, 4H), 3.75–3.39 (m, 20H), 1.13 (t, *J* = 6.9 Hz, 6H) ppm. ESI-MS: *m*/*z* calculated for C_80_H_68_N_4_O_10_ [M + H]^+^ 1244.5, found 1244.5.

### 2.2. Preparation of Perylene Diimide-Based Nanoparticles

The solution of PTA (2 mg) in THF (1 mL) was added dropwise to 5 mL of distilled water under continuous sonication (KQ5200B Ultrasonic machine from Kunshan Chaosheng Instrument Limited company, Suzhou, China). After 10 min, the organic solvent was removed by bubbling with nitrogen to obtain PTA-NPs as a homogeneous blue solution. The concentration of PTA-NPs was calculated using a standard curve of the UV/Vis absorbance of PTA [39].

### 2.3. Measurement of PCE

The PCE (η) of the PTA-NPs was calculated according to Equation (1).
(1)$$\eta =\frac{hA\left(T_{max}-T_{surr}\right)-Q_{Dis}}{I\left(1-10^{-A635}\right)},$$
where $T_{max}$ (°C) and $T_{surr}$ (°C) are the real-time temperature and surrounding temperature, respectively. A635 refers to the absorbance of the PTA-NP solution at 635 nm (1.25 W/cm^2^) under the experimental concentration. The value of *hA* was derived from Equation (2) [40].
(2)$$\tau_{s}=\frac{m_{D}C_{D}}{hA},$$
where $\tau_{s}$ is the time constant for heat transfer of the system, which was determined according to the Equations (3) and (4).
(3)$$T=-\tau_{s}\ln\left(\theta \right).$$
(4)$$\theta =\frac{T-T_{surr}}{T_{max}-T_{surr}}.$$

In Equation (2), $m_{D}$ and $C_{D}$ are the mass (3.0 g) and heat capacity (4.2 J·g^−1^) of the PBS (phosphate-buffered saline) used to disperse the NPs. In Equation (1), $Q_{Dis}$ represents the heat dissipation from the light absorbed by the water and the quartz sample cell. $Q_{Dis}$ was calculated according to Equation (5).
(5)$$Q_{Dis}=\frac{m_{D}C_{D}\left(T_{max\left(water\right)}-T_{surr}\right)}{\tau_{s}},$$
where $T_{max\left(water\right)}=35.9\;{}^{\circ}C$ (measured by thermal imager), $T_{surr}=26.9\;{}^{\circ}C$, and $\tau_{s\left(water\right)}=667.14$.

### 2.4. Detection of Reactive Oxygen Species (ROS) Generation 

To detect ROS generation with PTA-NPs, the nonfluorescent dichlorodihydrofluorescein (DCFH) was used as a probe, which can be converted to the highly fluorescent 2′,7′-dichlorofluorescein (DCF) in the presence of PTA-NPs under 635 nm laser irradiation. The fluorescence emission spectra of the DCF solution were measured within 500–588 nm upon excitation at 488 nm. The intracellular ROS generation was detected using the same method with dichlorodihydrofluorescein diacetate (DCFH-DA) as a probe. Specifically, the cancer cells were cultivated with DCFH-DA for 4 h and treated with PTA-NP solution. After laser irradiation at 635 nm for 5 min, the fluorescence images of cancer cells were observed by fluorescence microscopy (Zeiss, Oberkochen, Germany) [41].

### 2.5. Cytotoxicity Assay of PTA-NPs

First, 200 μL of PTA-NP solutions with different concentrations (0, 1, 5, 10, 15, and 20 μg/mL) were added into the plates where A549 cells (6 × 10^3^ cells per well) were cultivated. After incubation for 4 h, half of the A549 cells in the presence of PTA-NPs were irradiated with a 635 nm laser (1.5 W/cm^2^) for 3 min, and the other half were kept in the dark. All cancer cells were cultivated for another 5 h, treated with MTT solution at 37 °C for 4 h in 5% CO_2_, and then measured by a microplate reader at 570 nm. For the co-staining study, the cancer cells were cultivated with calcein AM and PI for 4 h and treated with PBS only, laser only, PTA-NPs only, or PTA-NPs + laser [42].

## 3. Results and Discussion

### 3.1. Synthesis of PTA and PTA-NPs

The synthetic procedures of PTA molecule and PTA-NPs are shown in Figure 1 and Appendix A (see Appendix A). PTA with a D-A-D structure was synthesized through a Sonigashira coupling reaction with electron donor TPA and perylene diimide as the starting materials. The amphiphilic organic molecule was then self-assembled into the nanoparticles (PTA-NPs) by the reprecipitation method. The structure of the target molecule was confirmed by mass and NMR spectra (see Appendix A).

### 3.2. Morphology and Particle Size

The morphology of PTA-NPs was studied by transmission electron microscopy (TEM), and the particle size was determined by dynamic light scattering (DLS). As shown in Figure 1A, a spherical morphology was observed for PTA-NPs with an average diameter around 200 nm, which was helpful for the accumulation of nanoparticles at the tumor site through the enhanced permeability and retention (EPR) effect. In addition, the DLS analysis further confirmed the uniform dispersion of PTA-NPs in PBS solution with an average size of 200 nm (Figure 1B). The zeta potential (ζ) of PTA-NPs was −10 mV, indicating the good stability of the aqueous solution (Figure 1C).

### 3.3. Photophysical Properties

As shown in Figure 1D, PTA exhibited a broad absorption spectrum in the UV/visible and near-infrared region (NIR) due to the large conjugated system. After self-assembly into PTA-NPs, the absorption spectrum was red-shifted and broadened to some extent, with a maximum absorption peak extension to 800 nm, probably ascribed to the π–π stacking of the conjugated structure in the nanoparticles. PTA-NPs were dispersed very well in water and did not show any precipitation after storage for more than 2 months (insert of Figure 1D), suggesting excellent colloidal stability. Because of the intramolecular electron transfer from donor to acceptor upon excitation, neither PTA nor PTA-NPs displayed any significant fluorescence (Figure 1E), which was beneficial for heat generation via a nonradiative route. 

### 3.4. Reactive Oxygen Species (ROS) Generation

As shown in Figure 1F, the probe of DCFH was oxidized to DCF by ROS generated from PTA-NPs upon 635 nm laser irradiation, and the fluorescence intensity of DCF increased linearly with time in the presence of PTA-NPs. In comparison, the probe of DCFH without PTA-NPs did not show any sign of fluorescence under the same conditions. Therefore, PTA-NPs upon laser irradiation could efficiently generate ROS for PDT applications.

### 3.5. Photothermal Properties

To evaluate the photothermal property of PTA-NPs, the temperature changes of PTA-NPs were recorded by a Flir-E6 thermal imager under 635 nm laser irradiation. First, the temperature elevation of PTA-NPs was recorded under laser irradiation of 635 nm at various laser power densities (0.5, 0.75, 1.0, 1.25, and 1.5 W/cm^2^). Figure 2A indicates that a higher laser power density resulted in a faster increase in the temperature of PTA-NPs. Moreover, the temperature enhancement of PTA-NPs exhibited concentration-dependent properties. As shown in Figure 2B, the temperature of PTA-NPs at the concentration of 45 μg/mL increased by 20 °C, while that at 135 μg/mL increased by 30 °C under 635 nm laser (1.25 W/cm^2^) irradiation for 600 s. In contrast, the temperature of DI water under the same conditions was not changed obviously. The excellent photostability of PTA-NPs was evaluated through five cycles of on/off laser irradiation (635 nm, 1.25 W/cm^2^) without obvious variation (Figure 2C). The PCE of PTA-NPs was calculated according to the single irradiation circulation (Figure 2D). From the linear curve of cooling time (t) vs. the negative natural logarithm of temperature (−ln θ) (Figure 2E), the time constant (τ_s_) was calculated to be 625 s. With these data in hand, the PCE was calculated as 43% according to the Equation (1). Meanwhile, the photothermal images of PTA-NPs at different concentrations were visually recorded through a thermal infrared imager after laser irradiation for 10 min (Figure 2F). Therefore, PTA-NPs could be used a potential photothermal agent for cancer phototherapy.

### 3.6. MTT Assay

The good biocompatibility of TPA-NPs was confirmed by MTT assay. As shown in Figure 3A, the viability of A549 cells was decreased to 78% with the concentration of TPA-NPs up to 20 μg/mL in the dark within 4 h. It should be noted that a similar viability of A549 cells was observed in the dark within 24 or 96 h (Appendix A). The results indicate a relatively low toxicity against cancer cells under dark conditions. The reason for the decreased viability is probably ascribed to the fluctuation of temperature during the experiment, the activity of cells, and the humidity of the environment. Furthermore, some of the cells would have inevitably died naturally in the culture medium with the incubating time. In contrast, the viability of cancer cells incubated with 20 μg/mL TPA-NPs was reduced to 9.6% after laser irradiation (635 nm, 1.5 W/cm^2^, 3 min). The half maximal inhibitory concentration (IC_50_) of 6.5 μg/mL indicated the high photocytotoxicity of PTA-NPs under light irradiation.

### 3.7. Intracellular Co-Staining Assay

The good biocompatibility and the photocytotoxicity of PTA-NPs were further verified in the live/dead cell co-staining test, where the live cells were stained with calcein AM (green fluorescence), while the dead cells were stained with propidium iodide (PI) (red fluorescence). As indicated in Figure 3B, the A549 cells treated with “PBS only”, “laser irradiation only”, or “PTA-NPs only” exhibited a green emission, indicating a negligible damage effect of bare PTA-NPs or laser irradiation alone. In contrast, the cancer cells showed a red emission after the treatment with PTA-NPs and 635 nm laser irradiation, suggesting that almost all of the cells were killed by the phototherapeutic effect of nanoagents.

### 3.8. Intracellular ROS Generation

HeLa cells incubated with PTA-NPs were investigated for intracellular ROS generation using DCFH-DA as a probe. A bright green emission was observed in the cytoplasm of A549 cells (Figure 3C) as a result of the ROS generation and the subsequent formation of green luminescent DCF. Thus, PTA-NPs could efficiently generate ROS in cancer cells under 635 nm laser irradiation.

## 4. Conclusions

In summary, organic nanoparticles (PTA-NPs) were prepared from a D-A-D structural organic molecule (PTA) through the nanoprecipitation method. PTA-NPs displayed good dispersibility and a uniform size in aqueous solution. The synergistic photothermal/photodynamic effects were demonstrated by the significant photothermal effect with PCE up to 43% and efficient ROS generation under 635 nm laser irradiation. The ROS generation of PTA-NPs in cancer cells was further evaluated using DCFH-DA as a probe. Lastly, the good biocompatibility and high photocytotoxicity of PTA-NPs were confirmed by MTT assay and a live/dead cell co-staining test. The results evidence that the as-prepared PTA-NPs can be used as promising nanomaterials in synergistic photodynamic and photothermal cancer therapy.

## Data Availability

The data presented in this study are available on request from the corresponding author.

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
