# Peer review of "Organic Nanoparticles Based on D-A-D Small Molecule: Self-Assembly, Photophysical Properties, and Synergistic Photodynamic/Photothermal Effects"

_materials, 2022, doi:10.3390/ma15020502_

Round 1
Reviewer 1 Report
Dear authors
In the paper a new material based on Perylene diimide has been designed for PTT and PD cancer therapy. The materials should be superior compared to others in literature. The paper could be of interest but at this stage is too preliminary.
Major consideration
The PTANPs should be compared to previously published similar NPs to understand the advantages.
The biological activity is underweight. In the MTT assay it could be seen that PTANP are toxic. There is a continuously decrease in viability although the time of exposure is very short. What happen after 96 hours?
Even in Figure 3B, what happen after long time? The images should be analyzed and quantified with histograms.
Fig3A and C lack of control. Please introduce controls as in Fig 3B
The paper lacks of discussion with the current literature.
Minor
English should be revised
Author Response
In the paper a new material based on Perylene diimide has been designed for PTT and PD cancer therapy. The materials should be superior compared to others in literature. The paper could be of interest but at this stage is too preliminary.
- The PTA-NPs should be compared to previously published similar NPs to understand the advantages.
Response: Thanks for the suggestion. To clarify the advantages of PTA-NPs, a Table 1 was provided in which the PTA-NPs were compared to those nanomaterials reported previously in the aspect of type, size, absorption and PCE.
Table 1. Comparison of different organic nanomaterials for PTT.
Nanomaterials |
Type |
Size |
Absorption (lmax) |
PCE |
Reference |
PDI-NPs |
Small molecule |
55 nm |
630 nm |
43% |
31 |
ZnP2 NPs |
Small molecule |
120 nm |
668 nm |
33% |
32 |
BAF4 NPs |
Small molecule |
79 nm |
1000 nm |
80% |
33 |
NDTB NPs |
Small molecule |
110 nm |
1050 nm |
40% |
34 |
- The biological activity is underweight. In the MTT assay it could be seen that PTANP are toxic. There is a continuously decrease in viability although the time of exposure is very short. What happen after 96 hours?
Response: Thanks for this suggestion. The results from MTT assay indicated that the synthesized materials might have a very low dark toxicity, probably ascribed to the fluctuation of temperature during the experiment, the activity of cells, and the humidity of the environment. However, all this variation was within the error range of the experiment. In fact, some of the cells will die naturally without any treatment in the culture medium along with the incubating time. The main purpose of the MTT assay is to verify the high phototoxicity of nanoparticles under laser irradiation, when compared with that of under dark condition.
- Even in Figure 3B, what happen after long time? The images should be analyzed and quantified with histograms.
Response: Thanks for this suggestion. The images with different colors (green: live cells; red: dead cells) can more directly reflect the survival of cells. The dark toxicity and photocytotoxicity of PTA-NPs were analyzed and quantified with histograms in MTT assay.
Reviewer 2 Report
The authors provided a well-written and scientifically sounding paper about the use of a type of organic-based NPs with PTT capabilities in cancer therapy. A few comments should be addressed before accepting the paper:
- In the introduction, the authors wrote "However, all these conventional cancer treatments could cause irreversible side effects to healthy cells in clinical application". State a few examples and add some statistics to strength the claims.
- In the introduction, the authors wrote "Up to now, a great number of photosensitizers including organic and inorganic functional materials have been designed and studied for PTT and PDT". Include a few examples of these chemicals from both categories, and include some references.
- The authors wrote "Inorganic materials generally exhibit high photother-mal conversion efficiency or efficient ROS generation". Include the reason behind this behavior.
- State what DPP-TPA stands for in the context of the provided reference.
- In Preparation of perylene-diimide-based nanoparticles, the authors should include the voltage and features of the ultrasonication approach, as well as the model of the used instrument and whether or not it was a pulsed application or a continuous approach.
- In Measurement of PCE, the authors should include the units of all parameters between brackets after the symbol or terms.
- Include the commercial manufacturer of the kit used to measure ROS or the provider of the chemicals.
- In ROS method, include whether or not controls were included and how the cells were handle before the measurement of the ROS activity (for instance, culturing of the cells, maintenance, subculturing and passaging and so on). These parameters are important to understand. Same comment about toxicity and cells. I would include a section just specifying how the cells were taken care of.
- What controls (positive and negative) were used for cytotoxicity? What dilution of MTT/MTS was done with media? Were the cells washed with PBS before addition of MTS? How were the NPs kept sterilized before addition to the cells? All these questions should be answered in the protocol to secure safety and reproducibility. Besides, how many times were the experiments repeated for toxicity and how many duplicated per experiment?
- A section about the calculations of statistical differences should be included specifying the software and mathematical models used.
- Since TEM images were taken, I would encourage the authors to measure the diameters of up to 100 particles in the images and provide a size using TEM (and compared to DLS), as it will be much more accurate.
- The authors mentioned "The zeta potential (ζ) of PTA-NPs was −10 mV, indicating the good stability in the aqueous solution". Explain why a negative zeta potential is synonymous of good stability.
- In Figure 1, a single NP was provided in the image. I would encourage the authors to provide an image with more NPs in the picture itself, as it shows clues about aggregation and concentration. Regarding this, what was the yield of the synthesis reaction?
- In results, the section reactive oxygen species (ROS) generation provides little discussion. Please, add some statements about ROS production via your NPs and how this might effect cancer cells and not healthy cells.
- In Figure 3, please, provide the Y-axis as cell concentration (cells/cm2 or cells/mL) instead of %, as this way does not show any real trends rather than cumulative absolute data. Also, provide statistical difference (t-student or similar) with asterisks and p values compared to the controls. Otherwise the graph does not tell anything.
- In MTT assay the authors should calculate IC50 values and compare those with others in literature for similar particles and cancer cell lines (as well as approaches).
- How were the NPs purified before used? Were they lyophilized?
As a minor comment, the grammar of the overall paper should be revised. Although there are not many typos, the overall style and soundness of some sentences is not well provided.
Author Response
The authors provided a well-written and scientifically sounding paper about the use of a type of organic-based NPs with PTT capabilities in cancer therapy. A few comments should be addressed before accepting the paper:
- In the introduction, the authors wrote "However, all these conventional cancer treatments could cause irreversible side effects to healthy cells in clinical application". State a few examples and add some statistics to strength the claims.
Response: Thanks for this suggestion. In the revised version, we have added some citations for the claims.
- In the introduction, the authors wrote "Up to now, a great number of photosensitizers including organic and inorganic functional materials have been designed and studied for PTT and PDT". Include a few examples of these chemicals from both categories, and include some references.
Response: Thanks for this suggestion. In the revised version, we have added some references related.
- The authors wrote "Inorganic materials generally exhibit high photother-mal conversion efficiency or efficient ROS generation". Include the reason behind this behavior.
Response: Thanks for this suggestion. In the revised version, we have added the reason behind this conclusion.
- State what DPP-TPA stands for in the context of the provided reference.
Response: Thanks for this suggestion. In the revised version, the meaning of DPP-TPA was explained according to the related reference.
- In Preparation of perylene-diimide-based nanoparticles, the authors should include the voltage and features of the ultrasonication approach, as well as the model of the used instrument and whether or not it was a pulsed application or a continuous approach.
Response: Thanks for this suggestion. In the revised version, the details mentioned were added.
- In Measurement of PCE, the authors should include the units of all parameters between brackets after the symbol or terms.
Response: Thanks for this suggestion. In the revised version, the units of all parameters after the symbol or terms were added.
- Include the commercial manufacturer of the kit used to measure ROS or the provider of the chemicals. In ROS method, include whether or not controls were included and how the cells were handle before the measurement of the ROS activity (for instance, culturing of the cells, maintenance, subculturing and passaging and so on). These parameters are important to understand. Same comment about toxicity and cells. I would include a section just specifying how the cells were taken care of.
Response: Thanks for this suggestion. Below is the details not included in the main text. Specifically, A549 cells (murine breast cancer cells) were all obtained from ATCC (Manassas, VA). These cells were cultured in DMEM containing 10% fetal bovine serum (FBS) in a 37 °C humidified incubator supplied with 5% CO2. For cellular ROS detection, after A549 cells were cultured in the medium for 24 hours, they were cultured in the medium containing PTA-NPs for another 4 hours, and blank A549 cells were used as a control. The A549 cells were then incubated with 10 µM DCFH-DA and DAPI for 30 min and washed three times with PBS. The cells were then irradiated with a 635 nm NIR laser for 5 min and observed under CLSM.
- What controls (positive and negative) were used for cytotoxicity? What dilution of MTT/MTS was done with media? Were the cells washed with PBS before addition of MTS? How were the NPs kept sterilized before addition to the cells? All these questions should be answered in the protocol to secure safety and reproducibility. Besides, how many times were the experiments repeated for toxicity and how many duplicated per experiment?
Response: Thanks for this suggestion. The positive control group had only nanomaterials but no laser irradiation, and the negative control group had no laser and no materials. MTT was prepared with PBS. This experiment did not use MTS, and only MTT was used. The cells were washed with PBS before adding MTT. The material was prepared with sterile water, and alcohol was sprayed during transfer and operation. The material will then be added to the cells in a sterile environment. The toxicity test was repeated three times, and each experiment was repeated 5 times.
- A section about the calculations of statistical differences should be included specifying the software and mathematical models used.
Response: Thanks for this suggestion. Generally, the statistical comparisons were made by ANOVA analysis and two-sample Student’s t-test.
- Since TEM images were taken, I would encourage the authors to measure the diameters of up to 100 particles in the images and provide a size using TEM (and compared to DLS), as it will be much more accurate.
Response: Thanks for this suggestion. We re-measured the TEM images of PTA-NPs. Unfortunately, we could not obtain an image with particles up to 100. As shown in Figure 1A, the average size of PTA-NPs was around 200 nm, which was almost consistent with the result from the DLS analysis (Figure 1B).
- The authors mentioned "The zeta potential (ζ) of PTA-NPs was −10 mV, indicating the good stability in the aqueous solution". Explain why a negative zeta potential is synonymous of good stability.
Response: Thanks for this suggestion. Nanoparticles with a highly negative potential exhibited the good stability in the aqueous solution because the repulsion between nanoparticles could reduce the aggregation of nanoparticles.
- In Figure 1, a single NP was provided in the image. I would encourage the authors to provide an image with more NPs in the picture itself, as it shows clues about aggregation and concentration. Regarding this, what was the yield of the synthesis reaction?
Response: Thanks for this suggestion. We re-measured the TEM images of PTA-NPs. Unfortunately, we could not obtain an idea image with much NPs. Moreover, the yield of the last coupling reaction is 60%.
- In results, the section reactive oxygen species (ROS) generation provides little discussion. Please, add some statements about ROS production via your NPs and how this might effect cancer cells and not healthy cells.
Response: Thanks for this suggestion. DAPI and DCFH-DA were used together as dyes for cells staining. DAPI can stain the nucleus with blue color, and DCFH can be oxidized by ROS to emit green fluorescence. From the green fluorescence in Figure 3c, we can confirm that PTA-NPs can produce reactive oxygen species in cells.
- In Figure 3, please, provide the Y-axis as cell concentration (cells/cm2 or cells/mL) instead of %, as this way does not show any real trends rather than cumulative absolute data. Also, provide statistical difference (t-student or similar) with asterisks and p values compared to the controls. Otherwise the graph does not tell anything.
Response: Thanks for this suggestion. As to the research about the photothermal cancer therapy. The dark and photocytotoxicity of photothermal agents in vitro were evaluated with cell viability (%) as Y-axis and nanoagents concentrations as X-axis, which were widely found in the previous literatures (Adv. Funct. Mater. 2017, 1605094; ACS Nano 2017, 11, 3797-3805; Nano Res. 2017, 10, 794-801; Nat. Commun. 2019, 10, 1192; ACS Biomater. Sci. Eng. 2020, 6, 5230-5239; Mater. Chem. Front., 2021, 5, 406-417)
- In MTT assay the authors should calculate IC50 values and compare those with others in literature for similar particles and cancer cell lines (as well as approaches).
Response: Thanks for this suggestion. The value of IC50 has been calculated and compared with the similar particles. The IC50 of PTA-NPs under laser irradiation was calculated to be 6.5 μg/mL.
- How were the NPs purified before used? Were they lyophilized?
Response: The crude solution of NPs was bubbled with N2 to remove the organic solvent (THF). After that, the obtained solution was filtered through a Millipore filter (pore size is 450 nm) to remove larger size particles.
- As a minor comment, the grammar of the overall paper should be revised. Although there are not many typos, the overall style and soundness of some sentences is not well provided.
Response: Thanks for the comment. The grammar, typos, and the soundness of the main text were checked and revised thoroughly.
Round 2
Reviewer 1 Report
Dear authors
Except for the first question “The PTANPs should be compared to previously published similar NPs to understand the advantages.” All the others were not adequately addressed. Please, check again the following suggestions.
The biological activity is underweight. In the MTT assay it could be seen that PTANP are toxic. There is a continuously decrease in viability although the time of exposure is very short. What happen after 96 hours?
Even in Figure 3B, what happen after long time? The images should be analyzed and quantified with histograms.
Fig3A and C lack of control. Please introduce controls as in Fig 3B
The paper lacks of discussion with the current literature.
Minor
English should be revised
Author Response
Please check the attachment below.

Reviewer 2 Report
The authors addressed most of the comments provided and now the paper can be moved forward in the publication line.
Author Response
Thanks.